# Leveraging social media in digital scholarship: Perspective from developing country students

**Theophilus Ocran[1], Kwaku Anhwere Barfi** [iD]**[1]\*, Kwame Kodua-Ntim[1], Paulina Nana Yaa Kwafoa[2], Christopher Kwame Filson[3]**

**1** Department of Information Technology & Research Support, University of Cape Coast, Cape Coast, Ghana, **2** Department of Client Service, University of Cape Coast, Cape Coast, Ghana, **3** Department of Collection Management & Technical Services, University of Cape Coast, Cape Coast, Ghana

\* kwaku.barfi@ucc.edu.gh

**Data Availability Statement:** The data are available from the UCCIRB for researchers who meet the criteria for access to confidential data: Osman Imoro (PhD), osman.imoro@ucc.edu.gh.

## Abstract

Albeit the increasing relevance of digital scholarship in contemporary educational settings, the onset of global pandemics like COVID-19 has necessitated the need for academic institutions to rely on social media for digital scholarship. Digital native students are leveraging on social media for digital scholarship to enhance communication and information dissemination. However, a study from higher institution in a developing country is missing from the global discussion on leveraging social media for digital scholarship. This study seeks to examine students' knowledge level in the use of social media for digital scholarship and the challenges associated with the use The study adopted stratified and non-probability voluntary response sampling methods because of the flexibility of these techniques. Data was collected from both undergraduate and postgraduate students of University of Cape Coast in Ghana. Students possess more than the average knowledge in social media for digital scholarship activities. However, students use of social media was for video presentations, online class, information sharing, publication of articles, search for academic related information, building proficiency in the search for information and making connections with individuals. Additionally, the conventional notion still holds that social media as a digital scholarship is susceptible to poor internet connection, jamming of digital systems and lack of adequate information on how to use digital scholarly platforms. The information literacy department of higher institutions are recommended to revise the content of their curriculum and incorporate mechanisms to leverage social media for digital scholarship to efficient disseminate scholarly outputs.

## Introduction

The inception of COVID-19 pandemic has not only demonstrated the need to pay attention to existing inequalities within populations and groups but also, how the Internet and digital technologies transform scholarly practices. As an idea which originated in the early 21st century, digital scholarship depends on institutional culture and organisation structure [1]. Digital scholarship is any scholarly activities that utilises opportunities for instruction and research created by the affordances of digital media [2]. It covers a wide range of activities but not limited to electronic submission of manuscripts, reassessment and evaluation of academic work

**Funding:** The authors received no funding for this research.

electronically and digital libraries [2]. Also, such activities include social media platforms due to its nature of openness, interactivity and sociability [3].

Social media is defined as a group of Internet-based applications that build on the technological foundations of Web 2.0, and that allow the creation and exchange of user generated content [4]. Specifically, social media platforms are considered as able to support a distributed and networked process of knowledge building and collaboration [5]. Academic institutions are working on finding innovative ways of engaging students through social media to mitigate the concerns of attrition rates. The widespread adoption and trends in scholars' use of social media for digital scholarship as an important tool for teaching and learning [6, 7].

Studies conducted by [8–12] on the use social media for digital scholarship shows that students are leveraging on technology to enhance scholarly communication. Whilst academic institutions continue to embrace technology, it is necessary to critically assess how university students as key constituent members appreciate such technologies [11]. Students can only appreciate these technologies when they have the needed knowledge level. According to [13], knowledge level is the theoretical or practical comprehension of a subject. They further indicated that knowledge level includes understanding facts, information, and abilities acquired through experience or study. The ability of the students to understand any information about digital scholarship constitute knowledge level. This aspect of the continuum, when unexplored will render the promise of taking full leverage of existing digital scholarship platforms equally ruined as some other exciting innovations also failed to meet expectations because they were erroneously premised on 'if we build it, they will come' [2].

Although, researchers have conducted studies on e-learning, m-learning, s-learning from the perspective of students or learners [14–16], this study will explore the use of social media for digital scholarship which is different from e-learning, m-learning, etc. In this study, digital scholarly are activities that make extensive use of one or more of the new possibilities for teaching and research opened up by unique affordances of digital media. These include, but are not limited to, multimedia, analysis of digital text, new forms of collaboration, new forms of publication new methods for visualising, technology-based database and analysing data [2].

Social media for digital scholarship is becoming a new tool for teaching and learning [2]. Moreover, a study with focus on students from higher institution in developing countries is missing from the global discussion on leveraging social media for digital scholarship. Inspite of the foregoing outlined issues, the use of social media for digital scholarship by students especially in Africa could be faced with myriad of challenges. This makes it difficult for students, lecturers, and other learners who seeks to benefit from the various services provided by digital scholarship to enjoy it full benefits. Some studies have examined the challenges students face in using social media as a key component of limiting digital scholarship [3, 17–19]. Although these studies investigated the use of social media for learning in a developing country [6, 9, 14, 20], there is no empirical evidence on the use of social media as a digital scholarship in Ghana. Again, this study wants to use the concept of Diffusion of Innovation (DOI) theory to understand how social media can be leveraged as a digital scholarship platform. The paper seeks to fill this gap. Findings from such studies justify the need to understand students use of social media within the context of a university setting in a developing country. Thus, the study sought to understand how social media is leveraged as a digital scholarship platform within a developing country for learning. The study sought to find out answers to the following questions:

1. What is the knowledge level of students in the use of social media as a digital scholarship platform?

2. What is the level of use of social media among students for digital scholarship activities?

3. What is the extent to which students' knowledge about social media predict their use as a digital scholarship platform?

4. What are the existing challenges associated with the use of social media as a digital scholarship platform among students?

## Literature review

### Theoretical framework

This study adopted Rogers diffusion of innovation theory which underpins the idea of four different features that inform the widespread use of technology [21]. The researchers adopted this theory to understand how social media as a technology is leveraged as a digital scholarship platform by students. Understanding these features will enable a more effective and efficient use of new technologies [22]. This theory underscore the use of technology features like compatibility, complexity, trialability and observability [21]. When social media is considered as an innovation, prior conditions like previous practices, individual needs, innovativeness and norms of the social system can also be associated with social media usage. As a result, individuals feel the need of using social media as a form of communication tool.

In the context of compatibility, having an Internet experience and being familiar with the Internet applications, provide the necessary conditions for the acceptance and widespread use of the technology [21]. Complexity is another effective feature that affects the acceptability of social media. [21] stated that an innovation, which does not require specific and complicated skills and understandings, will have the tendency to get a higher rate of adoption. Because of the simple and easy structure of social media, people do not need any complex technological skills. So this feature decreases the complexity level of this innovation and encourages people to try social media. In the view of [21], complexity and triability features have close relationship to each other. He argued that having low complexity level make people more enthusiastic to try new technological platforms. Also anyone who has a desire about social media can easily access these platforms without any prerequisites and this open access nature of social media is also important for the triability.

[23] mentioned that observability is one of the important motivation for the use of social media and in this context, the authors stated that people can use social media for conversation, interaction, and to feel part of a group to observe what is going on. In this sense, individuals can prefer to use social media for its communication, socialisation and self-expression benefits. If individuals make their decision as adaptation based on those or other reasons, they become users of social media platforms.

### Concept of digital scholarship

Universities are making major commitment to the social component of learning, devoting significant resources and effort to its promotion [1]. This gives academic libraries the opportunity to play a transformational role, moving from partners in generating knowledge to creators of knowledge. Digital scholarship encompasses a wide range of activities, most of which are not limited to using digital technology in teaching and research, but rather revolve around opening up values and ideologies in technology in order to better promote peer-to-peer networking and content sharing [7].

[2] characterised digital scholarship as constituting discovery (spearheaded by open data), integration (demonstrated by open publishing), application (which links up academia with the real world), and teaching (propelled by open education). In a similar vein, [24] identified five

unique perspectives from which digital scholarship is interpreted using teaching and learning, scholarly communication model, digital products, digital technologies, and research lifecycle. To them, the digital scholarship tools provide avenue to make intellectual work openly available to a wider audience.

## Forms and contents of digital scholarship services

To support and encourage digital scholarship research at the Universities, some libraries offers services such as online platform for teaching and learning, training room, consultancy services, digital resources, digital services and digital humanities [1]. Similarly, [2] indicated that digital scholarship services offered by academic libraries include research tools, research data services, digital technology support services, institutional repositories, digital scholarship seminars or trainings, and scholarly communication. In addition to the aforementioned, they indicated that audio, video, and image services as well as community education and online exhibitions are also some of the forms of digital scholarship services. Digital scholarship forms and services such as institutional repositories, electronic learning platforms (e-learning) among others are the facilities or resources that exploit digital scholarship.

## Actors in the adoption of digital scholarship platforms

[1] stressed that digital scholarship as a concept that is dependent on the culture of the institution and institutional organisation. Academic role in digital scholarship essentially looks at the way scholars leverage on digital facilities in other to increase their reputation and advance their scholarly endeavor [25]. Academic librarians practicing digital scholarship are positioned to act as partners to researchers and consultants on tools for digital publishing, digitisation and development of technologies to support and advance scholarship. Students who require assistance with digital projects and emerging technologies exude a certain level of confidence on the part of the librarian knowing how to use digital scholarship platforms [25].

The adoption of digital scholarship platforms in an institution ensures adequate storage and protection of institution's grey literature [2]. According to [26], this makes it possible for easy accessibility, widely disseminated and making it have a greater impact. Digital scholarship platforms have a new trend for institutions to communicate, collaborate, and disseminate their research [2]. Scholars can engage with the public and policymakers, and build their professional networks, enhancing their digital scholarship and impact.

## Social media in higher education

Social media is defined as group of Internet-based applications that allow the creation and exchange of user generated content [3]. [4] observed that the classification of social media includes blogs, social networking sites, virtual social worlds to mention but a few. Most students use social media to support their learning [5]. The usage of social media in digital scholarship has resulted in a networked for individuals pursuing intellectual objectives [8].

[27] share the view that social media platforms are transforming researchers' ability to contribute and collaborate across geographical and cultural fields in cutting-edge innovation. Students utilise social media to share knowledge and learning experiences with colleagues [22]. Though social media sites have been non-peer review platforms and mere appendices to digital scholarships, there are arguments in support of social media as a credible form of educational scholarship, particularly when social media use can be evaluated using the broader criteria for digital scholarships [2].

## Microblogging and its spread in academia

WhatsApp, Twitter, and Facebook are social mobilisation tools, but they also facilitate the dissemination of information [4]. According to [28], scholars use Twitter for a variety of purposes, including sharing practice-related information, requesting help and offering suggestions. [14] notes that Twitter's success comes from being a tool for social networking, characterised by "following" individuals with similar interests and being followed by others who share the same interests. The intrinsic value of microblogging is based on the network of connections established over time [29].

According to [30], Twitter is one of the most widely utilised social networking platforms among scholars to communicate and promote each other's work. Scholars utilise these microblogging technologies to interact with one other to stimulate innovative ideas [17]. According to [31], social networking platforms like Twitter are appropriate for public relations, academic organizations, scholarly associations and networks, as well as individual researchers in circulating information to information users. In similar vein, [18] revealed that Twitter is increasingly being used to support classroom, online, and work-based learning activities.

## Digital challenges in educational settings

Africa's internet penetration is nearing 40%, according to industry estimates, albeit it still trails below the global average [21]. Digitisation is enabling Africans to network, influencing their daily interactions and improving their access to various services such as education, financial and market services [22]. The use of social media for digital scholarship is expanding rapidly across Africa, according to Afrobarometer, although access is unequal [29].

[27] investigated digital scholarship practices in connection to the dimension of research and teaching among educational technologists and echoed how network participation now aligns and exceeds Boyer's paradigm for research, including persons rather than roles or organisations. A complex techno-culture system that is always evolving in response to internal and external stimuli, including lack of institutional policies, technical advancements and prevailing cultural values [20]. [32] also expressed concerns regarding the use of social media for scholarship, including privacy, blurring of boundaries, time constraints, content quality and risks connected to the rising dissemination of information. Other dangers highlighted by [2] were poor searching skills, plagiarism and copyright concerns.

## Methodology

The study adopted stratified and non-probability, voluntary response sampling methods because of the flexibility of these techniques [33]. To begin, the university was grouped into colleges using a stratified sampling approach. The study population was divided into five colleges: College of Agricultural and Natural Sciences, College of Health and Allied Sciences, Colleges are College of Education Studies, College of Distance Education, and College of Humanities and Legal Studies. This was done to ensure that the sample was representative of the study population. Social media announcements posted on students' WhatsApp platforms were used to find respondents. A 5-point Likert scale was used to assess the items in the survey.

Questionnaire was used to illicit responses from the students. Following an extensive review of literature, the questionnaire's questions were created. The researchers used expert panel analysis to review the developed questionnaires. Senior lectures at the Information Literacy Unit in UCC were used. The expert panel discussion was convened on 17[th] May 2021. All panel members were asked to undertake preparatory reading (systematic review of the questionnaire) before coming for the meeting. They made comments and suggestions on some

items in the questionnaire. After the corrections, the panel members finally approved the questionnaire for distribution to the selected participants. The researchers upheld the UCC's ethical standards. The UCC Institutional Review Board (UCCIRB) gave the researcher permission to conduct the study. Informed written consent was also obtained from all the participants of the study.

The research instrument was pretested on 20 students who participated in an annual e-resources training organised by UCC. However, students used in the pre-test were excluded from the actual study in order to minimise the contamination of data. A Cronbach alpha test was performed to ensure reliability of the instrument. The coefficient for knowledge level of students in the use of social media as a digital scholarship, level of use of social media among students, extent to which students' knowledge about social media predict their use of digital scholarship and challenges associated with the use of social media as a digital scholarship platform were found to be .756, .716, .796 and .778, respectively. Since the Cronbach coefficient was higher than 0.5, it means that the internal consistency of the items in the questionnaire was satisfactory [34].

## Data collection procedure

Since the study was an internet-based survey, the questionnaire was converted to Google Forms and the hypertext link to the questionnaire was shared with students through WhatsApp and all other electronic communication platforms available to students. Each respondent answered the questionnaire once since double entry could bias the result. The questionnaire was introduced with clear instructions given on its nature and voluntary participation was assured. A period of two months (June 2021 to August 2021) was used to collect the data. After this period, the hypertext link to the questionnaire was deactivated and all other late responses were not accepted. Out of a total of 4115 e-mails sent to students, 1043 responded.

## Statistical analyses

The data gathered was downloaded through Microsoft Excel and converted to SPSS to aid statistical analyses. Data gathered were computed and scored based on the interpretation of the researchers. Composite scores on the items on knowledge and use of social media were to aid parametric testing. Statistical analyses were conducted based on the nature of the research questions. The data retrieved from the field had normal distribution. Data on research question one was analysed using one-sample t-test since the aim was to establish the knowledge level of students on use of social media for digital scholarship. For the sake of the one-sample t-test, a standard (test value) of 37.5 was set by the researchers; hence a mean score below this standard was regarded as below average knowledge level whilst a score above this standard was interpreted as above average knowledge level. Data on research question two was analysed using mean of means to determine the student's use of social media in leveraging digital scholarship to promote scholarly output. Data on research question three was analysed using linear regression to examine the extent to which student's knowledge of social media predicted their use. Finally, means and standard deviations were applied in analysing data on research question four, which sought to investigate perceived challenges related to the use of social media in leveraging digital scholarship.

## Results

The descriptive statistics for the constructs are shown in Table 1.

From Table 1, the number of males involved in the study were 621 (59.5%) whiles their female counterparts were 422 (40.5%). This clearly indicated that males outnumbered females

**Table 1. Description of demographic characteristics of respondents.**

| Demographic | | Frequency | Percentage (%) |
|---|---|---|---|
| Gender | Male | 621 | 59.5 |
| | Female | 422 | 40.5 |
| Level of study | Level 100 | 279 | 26.7 |
| | Level 200 | 246 | 23.6 |
| | Level 300 | 196 | 18.8 |
| | Level 400 | 214 | 20.5 |
| | Post-graduate | 108 | 10.4 |
| College | Agriculture and natural sciences | 199 | 19.1 |
| | Education studies | 409 | 39.2 |
| | Health and allied Science | 117 | 11.2 |
| | Humanities and legal studies | 318 | 30.5 |

N = 1043

in the study however did not compromise the results of the study. In terms of level of study, level 100 students formed the majority with 279 (26.7%) followed by level 200 students who also constituted 246 (23.6%). The level 400 students made up of 214 (20.5%) of the total sampled students with 196 (18.8%) of the same sampled students being level 300 students. The group with the lowest representation were postgraduate students who constituted 108 (10.4%) of the total sampled students. The College of Education Studies had the highest number of student representation in this study, thus 409 (39.2%) with the College of Humanities and Legal Studies that followed next with 318 (30.5%) students. The College of Agriculture and Natural Sciences had a total number of 199 (19.1%) being represented in the study. The College of Health and Allied Sciences had the lowest representation, thus, 117 (11.2%) of students of the total sampled respondents.

## Research question 1: What is the knowledge level of students in the use of social media as a digital scholarship platform?

Research question one sought to examine the knowledge level of students in the use of social media as a digital scholarship platform using a 15-item scale. Composite score of the scale was calculated for each respondent to aid parametric testing. One sample t-test was used to analyse data on this research question. As stated earlier, a standard average (test value) of 37.5 was set, thus a mean below this set value was regarded as below average knowledge while a mean above this standard was interpreted as above average knowledge in the use of social media from the one sample t-test on students' knowledge in using social media Table 2.

From the results presented in Table 2, there is a significant difference in the standard value and mean of the participants in relation to their knowledge of social media $t$ (1042) = 35.86, $p$ < .001. Comparing the mean score of the respondents (M = 42.73) regarding their knowledge of social media to the test value (test value = 37.5) set by the researchers' interpretation, it

**Table 2. One sample t-test for students' knowledge in using social media.**

| | T | Df | Test value | Sig (2-tailed) | MD |
|---|---|---|---|---|---|
| Knowledge of Social Media | 35.86 | 1042 | 37.5 | .000 | 5.2 |

N = 1043, Mean = 42.73, SD = 12.46

could be deduced that students have above average knowledge level in the using social media as a critical element of digital scholarship.

## Research question 2: What is the use of social media among students for digital scholarship activities?

This research question investigated how students in the UCC use social media for digital scholarship activities.

**Test for normality.** The normality assumptions are fundamental to data presentation. According to [35], standard normal probability plots also called 'Normal P-P Plot' provides standard basis for testing assumptions. [35] further indicated that an observation of reasonable straight normal probability plot is an indication of normality. For this study, the result of the Normal P-P plot is reported in Fig 1.

It can be observed from Fig 1 that the line passes through a number of points suggesting a fairly straight line. The nature of the points or dots along the diagonal line indicates that the assumption of normality is met. Hence the study proceeds to use standard deviation in reporting the results.

To examine this, a 12-item scale was developed by the researchers the uses of social media among students. Data on this research question was analysed by conducting item analyses using means and standard deviation. The item with the highest mean was interpreted as the area that received the most use in leveraging social media as a digital scholarship platform. Results from the analysis of data on research question two is depicted in Table 3.

From the results in Table 3, students general use of social media (M = 2.73, SD = .96) was particularly aimed at for the purposes of discovery and information filtering (M = 3.23,

**Fig 1. Plot of Normal P-P.**

**Table 3. Uses of social media.**

| Items | M | SD |
|---|---|---|
| Research purposes | 2.91 | 1.099 |
| Search for academic related information | 2.74 | .971 |
| Video presentations, interview, and online class | 2.87 | 1.031 |
| Network analysis | 2.31 | .931 |
| Building proficiency and making connection | 2.69 | 1.062 |
| Data mining | 2.17 | 1.148 |
| Request support from appropriate individuals | 2.79 | 1.026 |
| Publication of articles | 2.80 | 1.085 |
| Scholarly and mobile communication | 2.37 | 1.092 |
| Information sharing | 2.81 | 1.062 |
| Discovery and information filtering | 3.23 | .908 |
| Identifying books and research publication | 3.13 | .918 |
| Total | 2.73 | .962 |

N = 1043, M = Mean, SD = Standard Deviation

SD = 9.08) and identifying book and research publication (M = 3.13, SD = .918). This was closely followed by the use of social media for research purposes (M = 2.91, SD = 1.099). Students most often used their social media exploits for video presentations, interview and online class (M = 2.87, SD = 1.03), information sharing (M = 2.81, SD = 1.06) and publication of articles (M = 2.80, SD = 1.08). Students also used social media platform to request support from appropriate individuals (M = 2.79, SD = 1.026), search for academic related information (M = 2.74, SD = 0.971), building proficiency in the search for information and making connections with other individuals (M = 2.69, SD = 1.062) and apply social media for scholarly and mobile communication within their circles (M = 2.37, SD = 1.092). In a contrary perspective, in terms of minimal use, students' lowest use of social media was for data mining (M = 2.17, SD = 1.148) followed by a lower use of social media that was geared towards network analysis (M = 2.31, SD = .931).

### Research question 3: How does students' knowledge of social media predict their usage?

The aim of research question three was to ascertain how students' knowledge in social media predicted their extent to use of social media platforms. The socio-technical approach underpins this study and explains the need for university students to have adequate knowledge to navigate the complex interaction that accompanies digital scholarship platforms (social media). The composite score of the results from students' knowledge and use of social media as computed. To analyse data on this research question, linear regression analysis was conducted, with students' knowledge on social media as the predictor and students use as the criterion variable. Results from the regression analysis is shown in Table 4.

A linear regression model was calculated to predict students use of social media based on their knowledge of the concept. Table 4 showed that a significant regression equation was found. The results indicate that $F(1, 1041) = 30.49$, $p < .01$, with an $R^2$ of .615. The results suggested student knowledge in social media ($\beta = .586$, $p < .01$) was a significant and positive predictor of student use of social media. The model explained 61.5% of the variance in usage of social media thus 62% of the variation in students' use of social media was predicted by their

**Table 4. Linear regression students' knowledge and use of social media.**

| Variables | B | R Squared ($R^2$) | SE B | β | t | P |
|---|---|---|---|---|---|---|
| Constant | 65.059 | .615 | 19.047 | | 3.41 | .000 |
| Knowledge in digital scholarship | 3.427 | | .262 | .586 | 13.08 | .000 |

$F = 30.49$, $df = (1, 1041)$, $N = 1043$

knowledge in social media. This meant that when students have adequate knowledge in social media, they were more likely to use it, whereas students with limited knowledge in social media were less likely to use it. This was because there was a significant and positive correlation between students' knowledge and use of social media.

### Research question 4: What are the challenges related to the use of social media among students?

The use of social media does come with its own peculiar challenges; thus, the purpose of this research question was to examine the various challenges students face in the use social media. A 10-item scale was developed by the researchers to examine these challenges. Data collected on this research question was analysed using means and standard deviations, thus the items with the highest means was regarded as the main challenges associated with the use of social media. Table 5 showed the results from the analyses of data on research question 4.

Table 5 showed the results on the various challenges of using social media. These challenges outlined are problematic for students in diverse ways. It was evident that the five major challenges were dominated by poor internet connection (M = 3.74. SD = .631), copyright concerns (M = 3.71, SD = .527), lack of support from appropriate staff (M = 3.63, SD = .536), content quality (M = 3.60, SD = .660) and lack of adequate information on how to use the platforms (M = 3.58, SD = .545).

### Discussion

The findings from this study have been discussed in relation to other studies that have been conducted on student's knowledge and use of social media for digital scholarship activities.

**Table 5. Challenges in using social media.**

| Items | M | SD |
|---|---|---|
| Unclear institutional policies | 3.56 | .502 |
| Lack of adequate information on how to use the platforms | 3.58 | .545 |
| Poor internet connection | 3.74 | .631 |
| Lack of support from appropriate staff | 3.63 | .536 |
| Dissemination of misinformation and rumours | 3.49 | .592 |
| Techno-culture and cultural values | 3.51 | .668 |
| Poor power supply | 3.50 | .527 |
| Content quality | 3.60 | .660 |
| Copyrights concerns | 3.71 | .527 |
| Privacy, Blurring of Boundaries and lack of credibility | 3.55 | .527 |

N = 1043

The discussion outlines where the findings from this study is consistent with previous research and areas where there are inconsistencies.

From the findings of this study, it appeared that students have adequate knowledge level and understanding of the nature of social media as a key component of digital scholarship. In fact, [26] put it succinctly that scholars are part of a complex techno-cultural system that is ever changing in response to both internal and external stimuli therefore they are expected to have an appreciable level of knowledge in the use of social media to match up dynamics in technological innovations, political and economic climates, and dominant cultural values. As a critical element of the key findings from this study, the above average level of knowledge implied that students have fair idea as to what social media entails, its nature as well as relevance to their academic work. This also support the concept of Rogers innovation theory which underpins the idea of four different features that inform the widespread use of technology. The knowledge level of students in the use of social media as a digital scholarship platform means that the students are compatible with this technology. Students having knowledge in the use of social media as a digital scholarship platform underscore their complexity level. Also, in the view of [21], complexity and triability features have close relationship to each other. This means that having high complexity level make people more enthusiastic to try new technological platforms like digital scholarship.

The dynamisms associated with social media usage required that users should have improved knowledge and understanding to gain the potential benefits it offers. Without adequate knowledge, it would prove difficult for students in any higher academic environment from either developed or developing to appropriately benefit from the system. Previous research by [36] also examined students' knowledge in digital scholarship and found that students have ample knowledge and information about the subject. The findings of [36], however, communicated the idea that students have adequate knowledge because digital scholarship activities are requirement for all students aside from the fact that students are naturally interested in using digital scholarship systems for learning. Inferring from the findings of this study and that of Watling [36] it was possible that students' level of knowledge in digital scholarship could influenced by a variety of factors including students' related factors as well as other factors outside the control of students. Since digital scholarship helps in better collaboration, faster communication, network opportunities, improvement for professional performances and new skillset for performance development [29], its relevance in academia cannot be understated. For this reason, it was imperative that students possess adequate knowledge in digital scholarship for more explorative ventures.

Another focus of this study was to examine students use social media for digital scholarship activities. The finding from this study indicated that students generally use some social media for digital scholarly activities. This finding contradicted and deviated from hitherto perception of cyber loafing activities among university students and employees [26]. Specifically, the findings revealed that students use social media platforms for digital scholarship activities such as creating personal learning environments, sharing content, upload presentation, supporting student review, feedback and evaluation. Identifying book and research publication. Other uses of social media are for research purposes, video presentations, interview and online class. Intuitively, knowledge-sharing element of networked scholar layers which subscribes to collective and distributed learning dimension focuses on the use of social media for knowledge dissemination among students in an academic environment. In the theoretical sense, findings from the study aligned the underpinnings of the phenomenological dimension of academic social media for digital scholarly activities which focused on individuals' relationships and the way they perceive the use of social media for academic purposes. In this sense, social media usage for digital scholarship activities is regarded as a third layer in the use of a scholar-

networked layer established and conceptualised by [26, 28]. For instance, [37], found that students' use of digital scholarship platforms for various purposes and these include research and article publication, search for relevant academic related information and analytics among other uses. This study offered an insight into some uses of social media and how students apply them in their academic work. Although empirical evidence from the findings of the study outlined the perspective of students in using social media platform, it must be emphasised that, there may be other potential uses of social media platforms that this study may not have outlined.

As part of the purpose of this study, focus was on whether student's knowledge of social media predicted their use of social media platforms. Findings from the regression analysis revealed that students who have adequate knowledge on social media were likely to use the various social media platforms that are available. Furthermore, 62% of the students' use of digital scholarship platforms was predicted by the knowledge students have in relation to social media. Deriving from the above, it could be implied that students with less knowledge in digital scholarship were likely to ignore the use of social media in their academic work. It was likely that the only time students with low knowledge in social media are when its use is made compulsory by lecturers and course instructors. In such situation, students may not understand and experience the various benefits associated with the use of social media platforms in their academic work. The above average usage level of social media provided the empirical basis for the usage component of techno-cultural layer where social media users in academia are perceived as agencies within a techno-cultural construct that distinguishes between implicit and explicit user participation.

[38] revealed that for students to appropriately use digital scholarship and other related technologies it is imperative that they have the maturity and knowledge to explore and relate to the various aspects of digital scholarship. [26] also stressed this point of view regarding the use and benefits from digital scholarship, they should be well equipped with the necessary knowledge, information, and skills to effectively learn, explore and improve their academic work.

Per the complex nature of digital scholarship, challenges are inevitable. From the findings of this study, major challenges of digital social media include poor internet connection, unclear institutional policies, copyright concerns, issues such as content quality, privacy and blurring of boundaries. It was important that networking component (infrastructure) is an indispensable element for efficient and proper usage of social media platforms. Again, lack of support from appropriate staff and lack of adequate information on how to use the platforms. These challenges were just a few of the various challenges students who use social media face. With challenges such as this, it made the use of social media stressful, unentertaining, and frustrating. A few researchers have also outline similar challenges in the wake of the inception of digital scholarship and technologies in academia. [25, 26] outlined various challenges of digital scholarship such as lack of appropriate digital resources, low acceptance among potential users, lack of assistance when users are in critical need. These challenges were evident from various empirical studies, but there was a chance that these challenges differ across contexts. In this regard, there was the need to examine and effectively tackle these challenges in order to make it user friendly. From a broader perspective, these challenges offer an opportunity to improve social media platforms to make it more accessible and available to students.

## Practical implications

For policy measures to contribute to efficient leverage of social media for digital scholarship for information sharing and scholarly output dissemination, academic institutions should take

some specific steps. The information literacy department of higher educational institutions should revise the content of the current curriculum and incorporate mechanisms to leverage digital scholarship platforms to efficient dissemination of digital scholarly output among academic peers. This also support the concept of Rogers diffusion innovation theory which underpins the idea of four different features that inform the widespread use of technology. Also, the information, communication and technology centres in universities should embark on an awareness campaign which seeks enlighten students on all available social media platforms that can be leveraged for efficient dissemination of digital scholarly works.

Similarly, institutions of higher learning most especially in developing countries should commit additional financial resources in providing internet connectivity as well collaborate with internet services providers (ISPs) to provide high-speed internet connectivity to facilitate the use of social media through digital scholarship platform. Whereas encouraging university students to make efforts to acquaint themselves with new and emerging trends in leveraging social media and other digital scholarship platforms, academic libraries should offer relevant information on how to make proper use of social media for quality information searches. This will build students' capacity to improve their academic and research work for scholarly output.

## Conclusion and recommendation

While research on the use of social media for digital scholarship activities mediated by social networking technologies is just beginning to emerge in most developing countries and has garnered some attention in recent years. A number of other empirical studies has analogous findings which align with this study. [39] found that university students are growing more familiar with and use of social networking platforms. A similar finding was reported by [2, 24, 28]. These researchers found that participants believed that social networking tools could bring advantages to local and global research collaboration. In spite of certain similarities, Research-Gate and Academia.edu, among other academic social networking sites, provide various opportunities for researchers to share research and network professionally. At the same time, sites like Zotero and Mendely provide its members special features that tend to encourage them to do particular things. Studies have clarified the importance of digital scholarship and how helpful it can be in diverse areas. From the finding of this study, it can be concluded that although university students have adequate knowledge of social media as a digital scholarship platform however their level of knowledge is not at a very high level. Meanwhile, a high level of knowledge about social media for digital scholarship activities is critical for students to use the various available social media platforms for learning. Just as it pertains in literature, social media for digital scholarship activities in developing countries have various associated challenges including problems with Internet access, lack of digital resource and lack of remote access of some university's social media handles. Academic institutions of higher learning ought to find sustainable and lasting solutions to these challenges that would go a long way to improve students' use of social media in leveraging digital scholarship.

For policy measures to contribute to efficient leverage of social media platforms for information sharing and scholarly output dissemination, the information literacy department of higher educational institutions should revise the content of the current curriculum and incorporate mechanisms to leverage digital scholarship platforms to efficient dissemination of scholarly output among academic peers. Also, the information, communication and technology centres in universities should embark on an awareness campaign which seeks enlighten students on all available social media platforms that can be leveraged for efficient dissemination of scholarly works.

## Limitations and suggestion for future works

The study should be perused within the context of its limitations among the self-report nature of the data. It was believed that students' responses might have been riddled with recall bias or inaccuracies, as is the case with most self-administered online questionnaires. In addition, the study was a snapshot of a period and not longitudinal research, it became difficult to observe the overall behavioural pattern. The use of multi-level sampling implies that any attempt at generalise the study must be made with caution since it was likely that the students who partook in this study might have possessed characteristics that were not proportionate to the general student population. Given these discrepancies, more research is needed to examine the relationship between perceived compatibility of using social media for digital scholarship and perceived academic usefulness.

## Acknowledgments

The authors express our sincere gratitude to all the first year students of University of Cape Coast who participated and supported by providing responses to the study.

## Author Contributions

**Conceptualization:** Theophilus Ocran, Kwaku Anhwere Barfi, Kwame Kodua-Ntim.

**Data curation:** Theophilus Ocran, Kwaku Anhwere Barfi.

**Formal analysis:** Theophilus Ocran, Paulina Nana Yaa Kwafoa.

**Investigation:** Theophilus Ocran, Kwaku Anhwere Barfi, Paulina Nana Yaa Kwafoa, Christopher Kwame Filson.

**Methodology:** Theophilus Ocran, Kwaku Anhwere Barfi, Kwame Kodua-Ntim, Christopher Kwame Filson.

**Project administration:** Kwaku Anhwere Barfi.

**Resources:** Theophilus Ocran.

**Software:** Theophilus Ocran.

**Supervision:** Paulina Nana Yaa Kwafoa.

**Validation:** Theophilus Ocran, Kwaku Anhwere Barfi.

**Writing – original draft:** Theophilus Ocran, Kwaku Anhwere Barfi, Paulina Nana Yaa Kwafoa.

**Writing – review & editing:** Theophilus Ocran, Kwaku Anhwere Barfi, Kwame Kodua-Ntim, Christopher Kwame Filson.

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
