## [Decision Letter · Decision Letter 0]

27 Mar 2023

PONE-D-22-28520Leveraging social media in digital scholarship: Perspective from developing country studentsPLOS ONE

Dear Dr. Barfi,

Thank you for submitting your manuscript to PLOS ONE. After careful consideration, we feel that it has merit but does not fully meet PLOS ONE’s publication criteria as it currently stands. Therefore, we invite you to submit a revised version of the manuscript that addresses the points raised during the review process.

Please note thatthe manuscript went through three rounds of reviews and the comments are attached for your perusal.you are to response to the comments from each of the reviewers and submit a revision of the manuscript for the next round of review.

We look forward to receiving your revised manuscript.

Kind regards,

Eric Amankwa, Ph.D.

Academic Editor

PLOS ONE

Journal Requirements:

Reviewers' comments:

Reviewer's Responses to Questions

**Comments to the Author**

1. Is the manuscript technically sound, and do the data support the conclusions?

Reviewer #1: Yes

Reviewer #2: No

Reviewer #3: Yes

2. Has the statistical analysis been performed appropriately and rigorously? 

Reviewer #1: Yes

Reviewer #2: Yes

Reviewer #3: Yes

3. Have the authors made all data underlying the findings in their manuscript fully available?

Reviewer #1: Yes

Reviewer #2: Yes

Reviewer #3: Yes

4. Is the manuscript presented in an intelligible fashion and written in standard English?

Reviewer #1: Yes

Reviewer #2: Yes

Reviewer #3: Yes

5. Review Comments to the Author

Reviewer #1: Thank you for submitting PLOS ONE. This is a well-written paper that covers an interesting topic. However, there are a few issues that the authors need to address.

1. The novelty of this paper is not clearly stated. The authors need to clearly highlight the main contributions of this study by highlighting the research problem(s), existing gaps in the literature, and how such gaps will be fulfilled.

2. The authors are advised to sperate the conclusion and recommendations into two sections: 1) practical implications (recommendations), and 2) conclusion.

3. Please avoid the use of bullets. It is suggested to present the recommendations in the form of discussion.

4. The authors are advised to update the literature review with more recent and related literature. This includes but not limited:

- Investigating the Impact of Social Media Use on Student’s Perception of Academic Performance in Higher Education: Evidence from Jordan. Doi: https://doi.org/10.28945/4661

- Social media usage and acceptance in higher education: A structural equation model.

Doi: https://doi.org/10.3389/feduc.2022.964456

Reviewer #2: REVIEWERS COMMENTS

Title: Leveraging social media in digital scholarship: Perspective from developing country students

A. General Comments to the Background:

1. The paper seeks to understand, from students’ perspective, Social Media (SM) use in Digital Scholarship (DS). The title could be reviewed to “Leveraging social media for digital scholarship: Perspective from developing country students”.

2. The abstract does not present clearly the research gap this paper seeks to address and the specific contributions it makes. In simple terms, what has been done so far, what has not been done and why authors think a developing country (DC) perspective is relevant.

3. This might have resulted from a lack of clear research objectives and questions in the main paper that relates to the overarching purpose: “Thus, the study sought to leverage social media as a digital scholarship platform within a developing country for learning.” p3.

4. Nevertheless, the purpose is phrased in a manner that suggests an experimental/action study else, the purpose should be modified to: “Thus, the study sought to understand how social media is leveraged as a digital scholarship platform within a developing country for learning.”

5. If this paper focused on the “learning” dimension of DS, the authors should have made a conscious effort to make a distinction between existing similar e-learning, m-learning, s-learning, etc, studies that has been done from the perspective of students/learners and this paper.

6. The research objectives and questions/hypothesis should flow from the overarching purpose. Whilst there are no explicit research questions, authors however responded to questions in the analysis section.

7. It is also unclear what research problem the stated objectives sought to address.

B. Literature Review and Theoretical foundation

8. The literature provided by the authors scantly focused on social media use in higher education and the general digital challenges associated with it. Again, because of an unclear research problem, the review seems to lack focus and had unrelated paragraphs. The authors could consider providing empirical studies on the main concept of DS and their evaluation of what is lacking. For example, authors could look at studies that focused on specific DS platforms, DS platform adoption and use by different actors in the DS ecosystem (ie researchers, teachers, publishers, publication editors/managers, students) to emphasise why a student perspective is relevant. Again, such a literature review should look at the geographical divide to make a DC study relevant.

9. The study also lacks the theoretical/conceptual rigour required of such a scholarly publication. Digital platform adoption and use has enjoyed a salad of theoretical approaches this study could have benefited from. There are theories for guiding studies of this nature that seeks to unearth enablers and constraints of SM use for DS. Authors may review and select appropriate ones (see TAM, institutional theory, Resource-Based theory, etc).

10. Concepts and key constructs were overly defined, or not defined at all. For example, Social media had been defined already on page 2. The definition on page 4 is therefore superfluous. Similarly, key constructs such as knowledge level, students use, etc were not defined.

11. Moreover, there is the lack of relevant citations in some cases. For example, Since SM is a core concept in this paper, there is the need to justify why SM is a DS platform. There must be authorities to back it or provide strong conceptualisation of why SM is a DS platform. For example, this statement, on page 2, is not well grounded: “One of such digital scholarship platforms popular among many individuals is social media or sometimes regarded as social networking tools due to its nature of openness, interactivity and sociability, which is considered as a powerful driver of change for teaching and learning practices by many authors and scholars”. Again, since this paragraph seeks to draw readers’ attention to the research objective the paper seeks to explore, it is important to provide appropriate citations to support this claim: “Whilst academic institutions continue to embrace innovations, it is necessary to critically assess how university students as key constituent members appreciate such new changes. This aspect of the continuum, when unexplored will render the promise of taking full leverage of existing digital scholarship platforms equally ruined as some other exciting innovations also failed to meet expectations because they were erroneously premised on ‘if we build it, they will come’. It is worth noting that studies that have examined challenges students face in using social media as a key component of digital scholarship are limited.” In addition, this categorical statement is made on page 4: “Most faculty use social media to support their teaching (Weller, 2011). Has anything change a decade after? It is a hanging statement that needs to be discussed or discarded.

C. Methodology

12. This study adopted a quantitative approach and the authors well justified the use of survey. Thus the method was appropriate. However, the authors introduced the research questions at this point and it is unclear how they are related and how they collectively address the research problem.

13. Consequent from the previous comments on the lack of clarity on the research problem/objectives/questions, It is difficult to follow the analysis and discussions.

14. Similarly it is difficult to relate the findings and conclusions to the research problem this study seeks to address.

D. Language and Formatting Issues

15. A number of language errors should be considered for correction through a thorough proofreading. For example on page 2, there is an omission of the word “of” in the phrase “..combination electronic means”. Similarly, on page 3, the word “to” should be deleted from the phrase “the challenges associated with the use to of social media as a digital scholarship platform among students.”

16. Again, there is the need for consistency in the use of the English language. There seems to be a mixture of US and UK English language spellings. For example, Organisation, conceptualisation, Digitisation are UK spellings whilst emphasize, recognized, mobilization etc are US spellings.

17. Authors should also pay attention to formatting issues. For example, there should be consistency with the spacing between the last paragraph and the next title. In some cases, there are no spaces whilst there is one space in other cases.

18. Authors should also be consistent with the spelling of key concepts. For example Social media and social-media are all used in the paper.

E. Publishability of the Paper

19. Based on the comments foregoing, this paper will need major corrections to make it appropriate for publication.

Reviewer #3: This is an interesting study and the authors have collected a unique dataset using cutting-edge methodology. The paper is generally well written and structured. However, in my opinion the paper has some limitations.

In the abstract, the authors stated that “The data was collected from undergraduate Students in a developing country...” Since the research was carried out in a single developing country, I suggest that the name of the country should be captured in the abstract.

In the abstract the authors stated that “The study revealed that students possess more than the average knowledge in social media, however, their use of social media was restricted to only few platforms”.. It is suggested that the authors show the related number of responded who uses a particular platform.

Further, the authors did not indicate the semantic and the expert panel analysis of the items (observed variables) used in the research. This is needed because the items were not adopted but were self-generated by the authors, hence it is important to show the condition for maintaining an item and the process that the items went through before it adoption.

On page 3, the authors stated that “It is worth noting that studies that have examined challenges students face in using social media as a key component of digital scholarship are limited” .. It is suggested that the authors give references/cite some researches that has tried to examine challenges students face in using social media.

The authors did not state whether they assumed normal distribution for the SD, this is significant in the sense that Standard deviation is based on the assumption that the data follows a normal distribution. If the data used is not normally distributed, then standard deviation may not be an appropriate measure to use. In such cases, other measures such as the interquartile range (IQR) or the median absolute deviation (MAD) may be more appropriate… The authors should consider this..

In the analysis the authors did not report the Standard Error (SE), this should be reported as it indicate the reliability of the mean.

It is recommended that the authors refer to these papers to enhance the Literature of the work:

George Veletsianos and Royce Kimmons, Networked Participatory Scholarship: Emergent techno-cultural pressures toward open and digital scholarship in online networks Computers & Education; February 2012, Volume 58, Issue 2, Pages 766-

Thoma B, Chan T, Benitez J, et al. Educational scholarship in the digital age: a scoping review and analysis of scholarly products. The Winnower 2014; 7: e141827.77297

Husain A, Repanshek Z, Singh M, et al. Consensus guidelines for digital scholarship in academic promotion. West J Emerg Med Integr Emerg Care with Popul Heal 2020; 21: 883–891.

Gruzd, Anatoliy, and Melissa Goertzen. 2013. “Wired Academia: Why Social Science Scholars Are Using Social Media.” In Hawaii International Conference on System Sciences. Available at http://ieeexplore.ieee.org/xpls/abs_all.jsp?arnumber= 6480244&tag=1. Accessed August 18, 2016.

Jamali, H. R., Nicholas, D., & Herman, E. (2016). Scholarly reputation in the digital age and the role of emerging platforms and mechanisms. Research Evaluation, 25(1), 37–49.

Jordan, K. (2014). Academics and their online networks: Exploring the role of academic social networking sites. First Monday, 19(1).

Maron, Nancy L., and K. Kirby Smith. 2008. “Current Models of Digital Scholarly Communication: Results of an Investigation Conducted by Ithaka for the Association of Research Libraries.” Association of Research Libraries. Available at www.arl.org/news/6/1148#7YL4SgrKUk

Manca, S., & Ranieri, M. (2017). Exploring Digital Scholarship. A Study on Use of Social Media for Scholarly Communication among Italian Academics. In Esposito A. (Ed.), Research 2.0 and the Impact of Digital Technologies on Scholarly Inquiry (pp. 116-141). Hershey, PA: IGI Global.

Scanlon, E. (2014). Scholarship in the digital age: Open educational resources, publication, and public engagement. British Journal of Educational Technology, 45(1), 12-23.

Cabrera D, Vartabedian BS, Spinner RJ, et al. More than likes and tweets: creating social media portfolios for academic promotion and tenure. J Grad Med Educ. 2017; 9(4):421-5

Lastly the manuscript contains some grammatical errors. I recommend that it is referred to an expert for review.

6. PLOS authors have the option to publish the peer review history of their article (what does this mean?). If published, this will include your full peer review and any attached files.

Reviewer #1: **Yes: **Ahmad Samed Al-Adwan

Reviewer #2: **Yes: **Dr. Mark-Oliver Kevor

Reviewer #3: No

---

## [Decision Letter · Decision Letter 1]

12 Jun 2023

PONE-D-22-28520R1Leveraging social media in digital scholarship: Perspective from developing country studentsPLOS ONE

Dear Dr. Barfi,

Thank you for submitting your manuscript to PLOS ONE. After careful consideration, we feel that it has merit but does not fully meet PLOS ONE’s publication criteria as it currently stands. Therefore, we invite you to submit a revised version of the manuscript that addresses the points raised during the review process.

All weaknesses identified by the reviewers should be addressed before you resubmit the manuscript for consideration.

We look forward to receiving your revised manuscript.

Kind regards,

Eric Amankwa, Ph.D.

Academic Editor

PLOS ONE

Reviewers' comments:

Reviewer's Responses to Questions

**Comments to the Author**

1. If the authors have adequately addressed your comments raised in a previous round of review and you feel that this manuscript is now acceptable for publication, you may indicate that here to bypass the “Comments to the Author” section, enter your conflict of interest statement in the “Confidential to Editor” section, and submit your "Accept" recommendation.

Reviewer #1: All comments have been addressed

Reviewer #2: (No Response)

Reviewer #3: (No Response)

2. Is the manuscript technically sound, and do the data support the conclusions?

Reviewer #1: Yes

Reviewer #2: Partly

Reviewer #3: No

3. Has the statistical analysis been performed appropriately and rigorously? 

Reviewer #1: Yes

Reviewer #2: I Don't Know

Reviewer #3: No

4. Have the authors made all data underlying the findings in their manuscript fully available?

Reviewer #1: Yes

Reviewer #2: No

Reviewer #3: No

5. Is the manuscript presented in an intelligible fashion and written in standard English?

Reviewer #1: Yes

Reviewer #2: Yes

Reviewer #3: Yes

6. Review Comments to the Author

Reviewer #1: Thank you for resubmitting the revised version of your paper. The authors have addressed the reviewers' comments. Accordingly, I am satisfied with the current version.

Reviewer #2: Re-Review

Title: Leveraging social media in digital scholarship: Perspective from developing country students

1. There has been some substantial revision to the original paper, For example:

a. The abstract has been improved.

b. The purpose has been clarified.

c. Key concepts have been explained.

d. The language has been improved

e. Appropriate citations have been provided

2. However there are still issues to consider. For example:

a. Whilst the paper adopts the Diffusion of Innovation (DOI) Theory, it is not clear how it has been used in this study [neither does it reflects in the Research Questions, Analysis or Discussions]. For instance, you may change the theory for a theory that captures your key constructs or modify the constructs entirely to reflect your selected theory.

b. Whilst the purpose has been clarified, there is the need for the research questions to be related and help contribute to the purpose.

3. Publishability:

The paper can be published upon the address of the highlighted concerns

Reviewer #3: The paper lacks the foundamental for scale development.

The questions used to solicit responses from particpants lacks the basics statistical foundation for scale development. Example the author stated that:

"To examine this, a 12-item scale was developed by the researchers the uses of social media

among students. Data on this research question was analysed by conducting item analyses using

means and standard deviation. The item with the highest mean was interpreted as the area that

received the most use in leveraging social media as a digital scholarship platform"

The 12 item-scale nedd to go through a "semantic analysis" , "Expert panel analysis" and validation before it can be used.

In my previous comment I ask about the I-CVI threshold that was set for the questions (items).

In my candid opinion this paper is not suitable for publication.

7. PLOS authors have the option to publish the peer review history of their article (what does this mean?). If published, this will include your full peer review and any attached files.

Reviewer #1: No

Reviewer #2: No

Reviewer #3: No

---

## [Author Response · Author response to Decision Letter 1]

20 Jun 2023

Reviewer 2 

Whilst the paper adopts the Diffusion of Innovation (DOI) Theory, it is not clear how it has been used in this study [neither does it reflects in the Research Questions, Analysis or Discussions]. For instance, you may change the theory for a theory that captures your key constructs or modify the constructs entirely to reflect your selected theory. 

The authors get your concern why the Diffusion of Innovation (DOI) theory does not reflect in the research questions. Research question one sought to examine the knowledge level of students in the use of social media as a digital scholarship. The DOI theory underscore the use of technology features like compatibility, complexity, trialability and observability which have some linkage with users’ knowledge level. Also, additional information have been provided on the discussion part to link thhe DOI theory to the study.

Whilst the purpose has been clarified, there is the need for the research questions to be related and help contribute to the purpose. 

This has been done.

Reviewer 3 

The paper lacks the fundamental for scale development. The questions used to solicit responses from participants lacks the basics statistical foundation for scale development. The 12 item-scale need to go through a "semantic analysis" , "Expert panel analysis" and validation before it can be used. 

This has been done.

In my previous comment I ask about the I-CVI threshold that was set for the questions (items). 

The authors have gave information on the Cronbach coefficient was higher than 0.5, it means that the internal consistency of the items in the questionnaire was satisfactory.

---

## [Decision Letter · Decision Letter 2]

29 Aug 2023

Leveraging social media in digital scholarship: Perspective from developing country students

PONE-D-22-28520R2

Dear Dr. Barfi,

We’re pleased to inform you that your manuscript has been judged scientifically suitable for publication and will be formally accepted for publication once it meets all outstanding technical requirements.

Kind regards,

Eric Amankwa, Ph.D.

Academic Editor

PLOS ONE

Additional Editor Comments (optional):

Reviewers' comments:

Reviewer's Responses to Questions

**Comments to the Author**

1. If the authors have adequately addressed your comments raised in a previous round of review and you feel that this manuscript is now acceptable for publication, you may indicate that here to bypass the “Comments to the Author” section, enter your conflict of interest statement in the “Confidential to Editor” section, and submit your "Accept" recommendation.

Reviewer #1: All comments have been addressed

Reviewer #2: All comments have been addressed

2. Is the manuscript technically sound, and do the data support the conclusions?

Reviewer #1: Yes

Reviewer #2: (No Response)

3. Has the statistical analysis been performed appropriately and rigorously? 

Reviewer #1: Yes

Reviewer #2: (No Response)

4. Have the authors made all data underlying the findings in their manuscript fully available?

Reviewer #1: Yes

Reviewer #2: (No Response)

5. Is the manuscript presented in an intelligible fashion and written in standard English?

Reviewer #1: Yes

Reviewer #2: (No Response)

6. Review Comments to the Author

Reviewer #1: Thank you for submitting the revised version of your paper. I am satisfied with the current version.

Reviewer #2: (No Response)

7. PLOS authors have the option to publish the peer review history of their article (what does this mean?). If published, this will include your full peer review and any attached files.

Reviewer #1: No

Reviewer #2: No

---

## [Editor Report · Acceptance letter]

15 Sep 2023

PONE-D-22-28520R2 

Leveraging social media in digital scholarship: Perspective from developing country students 

Dear Dr. Barfi:

I'm pleased to inform you that your manuscript has been deemed suitable for publication in PLOS ONE. Congratulations! Your manuscript is now with our production department. 

Kind regards, 

on behalf of

Dr. Eric Amankwa 

Academic Editor

PLOS ONE